# Assessment of an Ultrasound-Guided Longitudinal Approach to the Thoracic Erector Spinae Plane Block in Cat Cadavers: Description of Dye and Contrast Medium Distribution

**DOI:** 10.3390/ani15223311

**Published:** 2025-11-17

**Authors:** Sara Carrillo-Flores, Marta Soler, Francisco Gil, Gonzalo Polo-Paredes, Francisco G. Laredo, Amalia Agut, Eliseo Belda

**Affiliations:** 1Hospital Veterinario Universidad de Murcia, 30100 Murcia, Spain; sacarriflores@gmail.com (S.C.-F.); mtasoler@um.es (M.S.); gpolo@um.es (G.P.-P.); laredo@um.es (F.G.L.); amalia@um.es (A.A.); 2Departamento de Medicina y Cirugía Animal, Facultad de Veterinaria, Universidad de Murcia, 30100 Murcia, Spain; 3Departamento de Anatomía y Anatomía Patológica Comparada, Facultad de Veterinaria, Universidad de Murcia, 30100 Murcia, Spain; cano@um.es

**Keywords:** fascial block, feline, locoregional anaesthesia, erector spinae plane block, ultrasound-guided, spinal nerves

## Abstract

The erector spinae plane (ESP) block is an ultrasound-guided locoregional anaesthetic technique that primarily provides analgesia to the vertebral laminae, spinous process, epaxial musculature, and overlying skin. However, several clinical studies in humans have demonstrated the potential of the ESP block to provide analgesia for both thoracic and abdominal surgeries. While this block has been extensively studied in dogs, to the authors’ knowledge, this is the first cadaveric study of a thoracic approach to the ESP block in cats. The objective of this research was to evaluate the feasibility of an ultrasound-guided longitudinal approach to the thoracic ESP block in feline cadavers. Fifteen cat cadavers were used, three for an anatomical study, and twelve to assess the ultrasound-guided ESP blocks. A mixture of methylene blue, lidocaine, and iopromide (0.4 mL kg^−1^) was bilaterally administered at the level of the transverse process of the seventh thoracic vertebra (T7). Then, a computed tomography (CT) scan and anatomical dissection were performed to assess the distribution of the mixture. The CT images revealed the distribution of contrast medium withing the epaxial musculature and the middle layer of the thoracolumbar fascia. Anatomical dissection frequently showed staining of the dorsal branches of the thoracic spinal nerves. In some cadavers, the ventral branches of T7 and the sympathetic trunk were also stained. These findings support the feasibility of the thoracic ESP block in cats and its potential to provide analgesia for surgical procedures involving the dorsal thoracic area.

## 1. Introduction

The use of regional anaesthetic blocks in veterinary medicine has become increasingly widespread. These techniques present numerous advantages, including enhanced pain control, a reduction in opioid usage, improved haemodynamic stability, and enhanced recovery after surgery [1]. Regional anaesthesia has also been associated with a decrease in perianaesthetic mortality in dogs and cats [2]. Fascial blocks are ultrasound-guided techniques, in which local anaesthetic is administered in a fascial plane, allowing a wide distribution and desensitisation of extensive anatomical areas [3]. These techniques are gaining popularity due to their effectiveness in pain relief, high safety profile, and low complication risk [4].

The epaxial muscle group includes the medial transversospinalis system, which comprises the spinalis, semispinalis, multifidus, rotators, interspinalis, and intertransversarius muscles, as well as the longissimus and iliocostalis muscles. The erector spinae muscles complex, which is part of the epaxial musculature, consist of the spinalis, longissimus, and iliocostalis muscles surrounded by the superficial layer of the thoracolumbar fascia. These muscles are located on the dorsal aspect of the vertebrae and ribs [5]. In the erector spinae plane (ESP) block, a local anaesthetic is administered into the interfascial plane located between the erector spinae musculature and the transverse processes (TP) [5,6,7], the mammillary process [8,9], or the caudal aspect of the mammillary process [10] of the spine. This block primarily provides analgesia to the anatomical structures innervated by the dorsal branches (DB) of the spinal nerves, including the vertebral laminae, facet joints, epaxial musculature, and overlying skin [11].

In human medicine, the ESP block was initially employed for the management of chronic pain [6]. However, it is now commonly used to provide analgesia for spinal [12], thoracic, and abdominal surgeries, such as breast surgery, rib fracture repair, and splenectomy [13,14,15,16]. In veterinary medicine, cadaveric studies have described this block in the thoracic [7], thoracolumbar [9], and lumbar [8] regions in dogs, thoracic region in horses [17], and lumbar region in cats [18]. Its clinical application has been primarily focused on areas innervated by DB, including interventions related to intervertebral disc disease in dogs [19,20], dorsal spinous process ostectomy and desmotomy in horses [21,22], and lumbar spinal surgery in cats [23]. Moreover, this technique may be suitable for surgeries involving the excision of injection-site sarcomas, a relatively common pathology in cats.

Despite the widespread use of the ESP block, some controversy remains, as the mechanism by which it provides analgesia to the lateral and ventral walls of the abdomen and thorax, areas innervated by the ventral branches (VB) of the spinal nerves, has not yet been fully elucidated [8,9,24]. So far, only a few cadaveric studies have observed VB staining. Among them, two cadaveric studies in dogs [5,25], one in horses [17], and one in cats [18] reported this fact. Although no anatomical pathways allowing local anaesthetic to reach the VB have been described, in live patients the diffusion of local anaesthetics across fascial layers may help explain this potentially broader analgesic effect [4]. Additionally, the lymphatic distribution of local anaesthetic has also been proposed as a possible mechanism for this diffusion in pigs [26]. Finally, the needle may inadvertently penetrate the thoracolumbar fascia during block performance.

To the authors’ knowledge, only one cadaveric study [18] and one case report [23], both focusing in the lumbar region, have been published on ESP block in cats. However, anatomical studies evaluating the spread of contrast medium and dye in the thoracic ESP have not been conducted in cat cadavers. The objectives of this study were to investigate the anatomy of the epaxial musculature as well as the DB and VB of the spinal nerves at the thoracic level and to evaluate the feasibility of an ultrasound-guided thoracic longitudinal ESP block approach in cat cadavers.

## 2. Materials and Methods

This study was approved by the University of Murcia’s Biosecurity Committee in Experimentation (CBE 692/2025). All the cat cadavers included were either deceased or euthanized for reasons unrelated to this study. The cadavers were obtained through voluntary donations from owners to the Donation Program of the Veterinary Faculty of the University of Murcia. The donated cadavers were immediately frozen and thawed for 48 h at room temperature before use. Then, it was verified (X-rays) that the cadavers did not have any congenital or acquired anomalies in the spine and ribs.

### 2.1. Anatomical Study

Three cat cadavers were used in this phase. To identify the epaxial muscle anatomy as well as the DB and VB of the spinal nerves, from the first thoracic (T1) to the fourth lumbar vertebra, the cadavers were positioned in lateral recumbency. A skin incision was made using a dorsal approach to the spine from T1 to the sacrum. The skin was carefully dissected down to the linea alba. Subsequently, the following muscles were removed in this order: latissimus dorsi, trapezius, rhomboideus, serratus dorsalis thoracis, and serratus ventralis thoracis. At this stage, the external intercostal, iliocostalis thoracis, longissimus thoracis, thoracic spinal, and semispinalis muscles were exposed. Next, the thoracolumbar fascia, longissimus thoracis, and multifidus thoracis muscles were dissected to visualize the long rotator muscles and the DB of the spinal nerves. Afterwards, the thorax was opened, the ribs cut at the costochondral junctions, and the viscera removed. The internal intercostal muscles were dissected to reveal the VB of the spinal nerves. Finally, the thoracic sympathetic trunk was exposed. The same procedure was performed bilaterally. All the dissections were performed by S.C.-F. and F.G.

### 2.2. Ultrasound-Guided Technique

Twelve cat cadavers were used in this phase. The injections were performed bilaterally at the seventh thoracic vertebra (T7), using a linear ultrasound probe (3–13 Hz SL1543, MyLabGamma Vet, Esaote, Florence, Italy). The injectate consisted of a mixture of methylene blue (MB) (Pancrear Quimica, AppliChem, Castellar del Vallès, Spain) diluted in lidocaine (Lidocaine 2%, Braun Medical, S.A., Rubí, Spain) and iopromide (300 mg mL^−1^, UltraVist300, Bayer, Berlin, Germany). The final concentration of MB was 0.5% in a mixture 50:50 of lidocaine/iopromide.

The dorsal and ventral areas of the thoracic and abdominal regions were clipped, and the cadavers placed in sternal recumbency with the elbows stretched. The transducer was positioned over the thirteenth rib in a sagittal plane, and the ribs were counted (identified by their acoustic shadow with rounded shape) from caudal to cranial (Figure 1). After identifying the seventh rib, the probe was directed dorsally until the TP of T7 appeared as an irregular hyperechoic structure, resembling the side of an armchair and producing acoustic shadowing [7]. Then, a sonovisible needle (Ultraplex 360°, 20 G, 100 mm, 30°, Braun, Melsungen, Germany) was introduced “in plane” from the caudal to cranial direction (Figure 2). The visualization of the needle in its whole path was registered in a binary category (yes or no). Once the tip of the needle was in contact with the dorsal aspect of TP, a small amount of the injectate (0.1 mL) was administered. When the spread of the injectate was judged adequate (anechoic pocket between the longissimus dorsi muscle and the TP of T7), a total volume of 0.4 mL kg^−1^ was administered. All ultrasound-guided injections were performed by the same operator (S.C.-F.).

### 2.3. Computed Tomography (CT) Study

A helical CT scan was performed from the sixth cervical (C6) to the thirteenth thoracic (T13) vertebra using a 16-slice multidetector CT scanner (Revolution, General Electric Healthcare, Madrid, Spain) 15–20 min after the injections. The cadavers were placed in dorsal recumbency, with the limbs extended along the longitudinal axis. The collimator pitch was adjusted to 1, with a slice thickness of 2.5 mm. A reconstruction interval with a 1.25 mm overlap was used. The kVp and mA were both set to 120. Standard reconstruction algorithms for bone and soft tissue were applied. Two specialists in imaging reviewed the reconstructed images and studied the distribution of the contrast (M.S. and A.A).

### 2.4. Spread Study

Immediately after the CT scan, the anatomical dissection of the cadavers was performed as described above. The nerves were classified as “stained” (nerves stained for 1 cm or more in their total circumference), “partially stained” (nerves stained less than 1 cm), or “not stained” [27]. The same assessment was conducted on the sympathetic trunk. The dissections were carried out by the same researchers (S.C.-F. and F.G.).

### 2.5. Statistical Analysis

A statistical descriptive analysis was conducted utilizing Microsoft Excel 365 software (Microsoft Corporation, Redmond, WA, USA) in conjunction with the Real Statistics Resource Pack plug-in (version 9.4.5, Copyright 2013–2021, Charles Zaiontz, www.real-statistics.com (accessed on 25 July 2025). The Shapiro–Wilk test was employed to assess the distribution of the data. The results are presented as the median (range) for data exhibiting non-normal distribution and as the percentage for categorical variables.

## 3. Results

Fifteen European shorthair cat cadavers were evaluated and included in the study.

### 3.1. Anatomical Study

Three cat cadavers, two males and one female, with a weight of 3.8 (1.98–4.38) kg, were used in this phase. After skin removal, the latissimus dorsi muscle and the superficial layer of the thoracolumbar fascia (origin of the latissimus dorsi muscle) were observed. The superficial layer of the thoracolumbar fascia covered the dorsolateral aspect of the epaxial musculature in the dorsal region. The thoracic portion of the trapezius muscle, partially fused with the latissimus dorsi, was also visible. Once the thoracolumbar fascia was detached and removed, the spinalis, semispinalis, longissimus, and iliocostalis muscles were exposed. The DB of the spinal nerves were then observed attached to the longissimus muscle after it was separated from the iliocostalis (Figure 3). The dissection of these muscles revealed the long rotator and the multifidus muscles, as well as part of the spinous processes of the thoracic vertebrae. The lateral and medial branches of the DB could not be identified. After removing the serratus ventralis thoracis and the intercostal muscles, the VB of the spinal nerves were observed emerging from the intervertebral foramen and extended ventrally to the caudal aspect of the ribs. Once the thoracic viscera were removed, the sympathetic trunk was observed alongside the aorta (Figure 3).

### 3.2. Ultrasound-Guided Technique

The injectate was administered to twelve cat cadavers (24 sides), including four females and eight males, with a body weight of 3.09 (1.6–6.98) kg. Needle visualization was considered adequate in all the injections. During the administration of the aforementioned mixture, an anechoic pocket was noticed in 20 out of 24 sides (83.33%) (Figure 4). A certain amount of the injectate was observed to flow back along the needle path on 4 out 24 injections (16.66%). Furthermore, part of the mixture was distributed intramuscularly in 18 out of 24 sides (75%). The distribution of the injectate was cranial in 7 out of 24 (29.16%), caudal in 6 out of 24 (25%), and bidirectional in 11 out of 24 (45.83%) sides.

### 3.3. Computed Tomography (CT) Study

The contrast medium was visualized in the target area in all cadavers, although in 2 out of 24 sides the epaxial distributions of the contrast medium were minimal. The distribution of iopromide was observed along eight (4–11) vertebral bodies, always between T1 and T13. The most frequent distribution intervals were T5-T12, T6-T11, and T1-T11, each of which occurred in 3 out of 24 sides (12.5%). Some amount of intramuscular distribution of iopromide was shown in all sides. The maximum cranio-caudal extension was eleven vertebral bodies. A ventral spread was observed through the seventh intercostal space in 17 out of 24 sides (70.83%) (Figure 5). In addition, contrast medium was also observed within the thoracic paravertebral space in 5 out 24 sides (20.83%).

### 3.4. Spread Study

The MB was observed in the epaxial musculature in 22 out of 24 sides (91.66%). Among them, 15 out of 22 sides also exhibited the seventh intercostal space stained. In the remaining 2 out of 24 sides, the dye was only found in the seventh intercostal space.

The DB were stained in 15 out of 24 sides (62.5%), with a median of 2 (0–9). The DB most frequently dyed was T7 (58.3%), followed by T6 (50%) and T8 (45.83%). The VB were observed dyed with a median of 0.5 (0–1). Among them, T7 was the only one stained in 12 out of 24 sides (50%). Finally, the sympathetic trunk was stained in 7 out of 24 sides (29.16%), with three of them considered partially dyed (Figure 6 and Figure 7). Staining of the sympathetic trunk was observed along one vertebral body in four out of seven sides. The maximum and minimum length dyed was three and half vertebral bodies, respectively. In all the sides where the sympathetic trunk was found stained, the VB of T7 was also involved.

## 4. Discussion

The present study showed that the ultrasound-guided administration of 0.4 mL kg^−1^ of a mixture of MB, lidocaine, and iopromide at the level of the TP of T7 resulted in a distribution of the injectate along the epaxial and intercostal muscles in cat cadavers. This technique frequently resulted in staining of several DB and one VB of the spinal nerves, as well as occasionally the sympathetic trunk, supporting the suitability of the thoracic ESP block in felines.

Anatomical studies describing the dorsal branches (DB) of the spinal nerves in cats are scarce. It has been previously reported that, in felines, the DB of the spinal nerves rapidly divide into medial and lateral branches [28]. As also occurs in humans and dogs, the medial DB provide sensory innervation to the epaxial muscles, vertebral laminae, and facet joints, whereas the lateral DB innervate the skin on the dorsolateral aspect of the back [29,30]. The above indicates that the ESP block could be used in surgeries performed in the epaxial region. However, the mechanism by which the ESP block could provide analgesia in areas innervated by the VB and sympathetic trunk is not fully understood. Within our anatomical study, we did not find any anatomical pathway between the thoracic epaxial and hypaxial musculature. This agrees with the results of Nobre et al. (2025) within the lumbar region in feline cadavers [18], as well as with previous studies in dog cadavers [7,8,9]. Some theories in humans suggest that perforations in the thoracolumbar fascia and intertransverse connective tissue, created by the DB of the spinal nerves and vessels, as well as by joints and ligaments, may serve as areas of low resistance, facilitating the spread of fluids [31,32,33]. The diffusion of the injectate through the connective tissue reaching the paravertebral space should also be considered [4]. Otero et al. (2020) detected dye in thoracic lymph nodes following ESP block at the sixth thoracic vertebra in a porcine living model. Systemic absorption and distribution of the local anaesthetic via lymphatic drainage should also be considered as a potential route of analgesic action [26]. Previous studies have reported the spread of contrast medium into the epidural space, which should likewise be regarded as a possible mechanism contributing to analgesia [8,25]. Finally, inadvertent perforation of the thoracolumbar fascia during ESP block approach may provide a plausible explanation for the analgesia observed in abdominal and thoracic surgeries.

As previously reported [5,7,8], the use of a linear ultrasound probe provides clear and well-defined images of the anatomical structures to perform the ESP block. It must be considered that the size of the transducer could be a limitation when ultrasound-guided regional anaesthetic blocks are implemented in small patients. In the present study, T7 was selected as a target vertebra because it is sufficiently from the scapula, which could otherwise hinder proper transducer placement. Therefore, in the longitudinal approach of the thoracic ESP block focused at T7, the absence of bony structures in the epaxial musculature allowed the use of a linear probe, despite the small size of the feline cadavers. The dorsolateral edge of the TP is considered a key landmark to perform the thoracic ESP block in dogs [7,25]. Moreover, the ultrasound images of the TP at T7 obtained in our study are in agreement with the shape of an armchair described by Portela et al. (2020) in dog cadavers [7]. Hence, the same morphology of the TP can serve as a reliable landmark for performing the longitudinal thoracic ESP block in cats and allows for comparation with previous studies.

In our study, the ultrasound-guided longitudinal approach previously described by Portela et al. (2020) in dog cadavers was performed [7]. Herrera-Linares et al. (2024) reported comparable outcomes between longitudinal and transversal approaches in the thoracic region [12,25] of dog cadavers. However, the transversal approach has been considered more appropriate in both the thoracolumbar [11] and lumbar [12] regions. Further comparative studies in cats should be necessary to elucidate whether the transversal approach could increase the amount of DB stained in the thoracic region.

A volume of 0.4 mL kg^−1^ per side was selected because it corresponds to a dose of 2 mg kg^−1^ of 0.5% bupivacaine or ropivacaine in a clinical scenario. Administration of a larger volume could exceed the recommended doses of these two commonly used local anaesthetics in cats [34,35,36]. Fascial blocks are inherently volume-dependent techniques in which the use of higher volume is common. Dilution of local anaesthetics is a commonly used alternative when a larger volume of injectate is required. However, it should be noted that higher dilutions are associated with a shorter period of action and less intense analgesia. Another important consideration is the density of the mixture of MB, lidocaine, and iopromide compared to that of a local anaesthetic alone. The density of the injected material has been directly associated with its distribution and diffusion through the connective tissue that makes up a fascia [4]. In the present study, lidocaine was added to achieve a density as close as possible to that of the local anaesthetics. However, it must be considered that both iopromide and MB modify the density of the injectate and consequently its spread and diffusion. Because of the limited injectate volume (0.1 mL) used to confirm needle tip placement and the large dimension of the erector spinae fascial plane, visualization of the anechoic pocket was not achieved in four sides. The decision to administer the full volume, in order to minimize the loss of the mixture, was then based on the needle trajectory and its contact with the TP of T7.

Previous studies in dog cadavers suggest that the ESP block could be an effective regional anaesthetic technique for providing analgesia to anatomical structures innervated by the lateral and medial DB [7,9,25]. In addition, clinical studies support the use of this block in spinal surgeries such as hemilaminectomies in both canine [19,20,37,38] and feline species [23]. Ferreria et al. (2019) reported that two (1–3) thoracic DB were stained with a volume of 0.5 mL kg^−1^ in dog cadavers making no distinction between lateral and medial DB [5]. Although, in the lumbar region, Nobre et al. (2025) administered two different volumes of 0.6 mL kg^−1^ and 0.4 mL kg^−1^, resulting in 4.5 ± 1.2 and 2.8 ± 1.3 DB stained, respectively, in cat cadavers [18]. These results are consistent with our findings, in which two (0–9) DB were stained following administration of 0.4 mL kg^−1^ of injectate. Portela et al. (2020) were able to differentiate medial and lateral DB in dog cadavers. These authors reported a higher distribution of injectate affecting four medial DB and four–five lateral DB, administering two different volumes of 0.3 and 0.6 mL kg^−1^ [7]. Similarly, Herrera-Linares et al. (2024) in two different approaches (longitudinal and transversal), with a volume of 0.6 mL kg^−1^, found a distribution of three–four medial DB and three lateral DB in dog cadavers [25]. In the present study, the small size of our specimens prevented differentiation between the lateral and medial DB. This finding is consistent with the observations of Nobre et al. (2025) [18], who also reported that such differentiation was not possible in cats, due their small size. This is an important aspect to consider, as the medial and lateral DB may have been stained in our cat cadavers to a larger extent than the main DB, potentially increasing the success rate of the block in a clinical setting. The different number of DBs stained among studies may be due to the injection volume administered, anatomical variations related to the target vertebra, anatomical differences among species, and also the physicochemical properties of the injectates employed.

It is important to highlight that the VB stained, as well as the intercostal distribution of contrast medium observed in our CT study, was only at T7. This supports the theory that perforation of the thoracolumbar fascia, occurring when the needle contacted the TP, may be the primary mechanism underlying the ventral distribution of the injectate. This finding is in agreement with Herrera-Linares et al. (2024), who reported one (0–3) and one (0–4) stained VB with the transversal and longitudinal approaches, respectively, in dog cadavers [25]. Similarly, Ferrerira et al. (2019) observed one VB (one out of eight injections) stained in their high-volume group in canine cadavers [5]. Delgado et al. (2021) also found staining of three VB (3 out of 14 injections) in adult horse cadavers, in which the injectate was administered at the level of the sixteenth thoracic vertebra [17]. Nobre et al. (2025) reported staining of the VB in the lumbar region of cat cadavers [18]. Although our findings show similar results, direct comparison should be made with caution, as the anatomical regions and even species investigated differ among studies.

Both anatomical dissection and CT studies revealed that the sympathetic trunk was reached by the injectate in some of our cat cadavers. This fact has not been previously reported. The involvement of the VB and sympathetic trunk suggests the capability of this block in providing analgesia to the thoracic wall and viscera. These results align with clinical studies in humans, which have demonstrated the effectiveness of this block in providing analgesia for thoracic and abdominal procedures [39,40,41,42]. To the authors’ knowledge, only two clinical studies in dogs describe the use of the ESP block for procedures other than spinal surgery, including sternotomy [43] and acute pancreatitis [44]. No clinical studies have been conducted in cats. Additionally, previous studies have reported the spread of contrast medium into the epidural space [8,25]. Nobre et al. (2025) did not observe epidural distribution of dye in cat cadavers [18]. In agreement with their results, we did not visualise contrast medium in the epidural space in the CT images. That is the reason why the vertebral canal was not open during our anatomical dissections.

The limitations of this study include the small number of specimens employed, which may influence the results. Additionally, the study was conducted on cadavers, which could affect dye spread due to the absence of physiological factors such as perfusion, lymphatic drainage, muscle tone, and ventilation. In addition, the frozen and thawed processes may alter the tissue properties, modifying the distribution of the injectate. It is also important to remark again that, despite the addition of lidocaine, the use of iopromide and MB modified the density and viscosity of the injected material when compared with local anaesthetics, which may potentially result in a different spread pattern. Moreover, it has been reported that the use of MB could overestimate the amount and the distance of target structures stained, due to its high capacity of diffusion and penetration into biological tissues [45]. Another consideration is that the dissection of the DB and VB involved significant tissue alteration, which may have modified anatomical barriers and led to an increased dye spread, as reported previously [7,46]. Finally, the dissections were performed by unblinded personnel, which may introduce a certain bias in the interpretation of the findings.

## 5. Conclusions

The ultrasound-guided longitudinal thoracic ESP block in cats is a feasible technique that achieves effective distribution of the injected fluid through both the epaxial musculature and the dorsal branches of the spinal nerves. This fact potentially makes regional anaesthetic block an effective method for providing analgesia to the dorsal area innervated by the dorsal branches of the spinal nerves. Further studies are needed to confirm its efficacy in providing analgesia to structures related to the ventral branches and sympathetic trunk.

## Figures and Tables

**Figure 1 animals-15-03311-f001:**
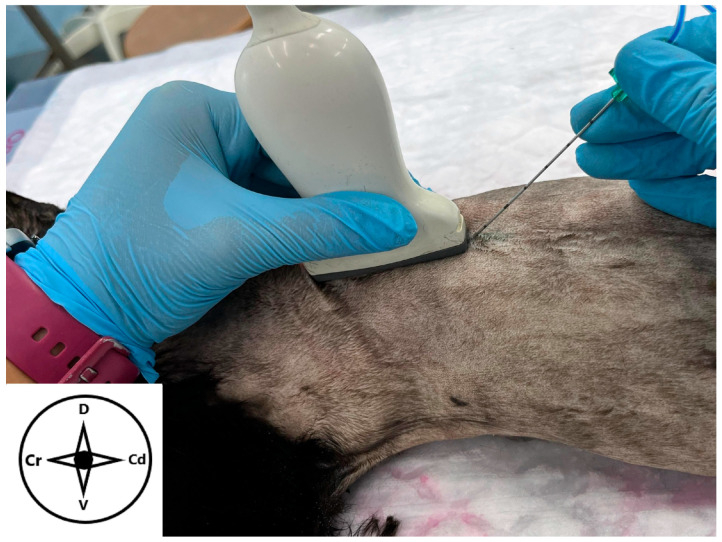
Position of the ultrasound probe to perform the longitudinal approach of the erector spinae plane block in a cat cadaver. The probe is placed dorsally over the spine at the level of the transverse process of the seventh thoracic vertebra. The needle is advanced “in plane” in a caudocranial direction. Cr: cranial, Cd: caudal, D: dorsal, V: ventral.

**Figure 2 animals-15-03311-f002:**
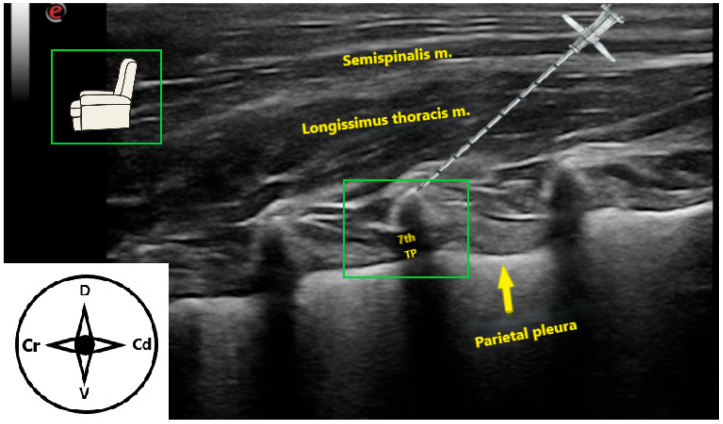
Ultrasonographic image of the epaxial musculature and reference points to perform a longitudinal approach of the erector spinae plane block at the level of the transverse process of the seventh thoracic vertebra in a cat cadaver. The transducer is orientated longitudinally in a parasagittal position, and the needle trajectory is shown in a caudocranial direction. TP: transverse process. Cr: cranial, Cd: caudal, D: dorsal, V: ventral.

**Figure 3 animals-15-03311-f003:**
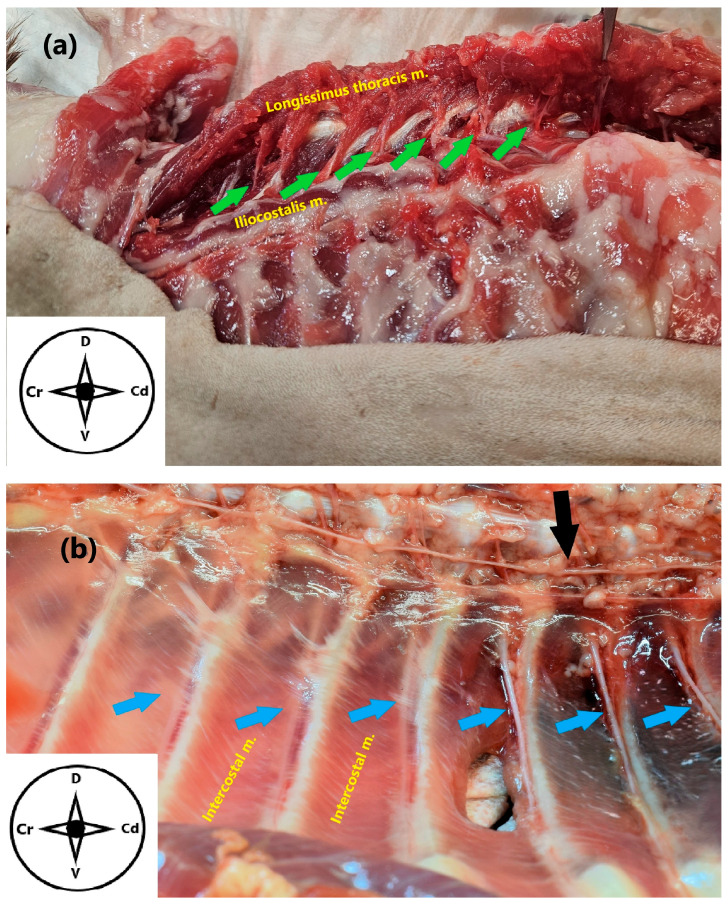
(**a**) Anatomical dissection of the epaxial musculature of the spine in a cat cadaver. The dorsal branches T2–T11 are shown (green arrows). (**b**) Zenithal view of the inner dorsal thoracic wall after anatomical dissection. The ventral branches T6–T11 (blue arrows) and the sympathetic trunk (black arrow) are exposed. Cr: cranial, Cd: caudal, D: dorsal, V: ventral.

**Figure 4 animals-15-03311-f004:**
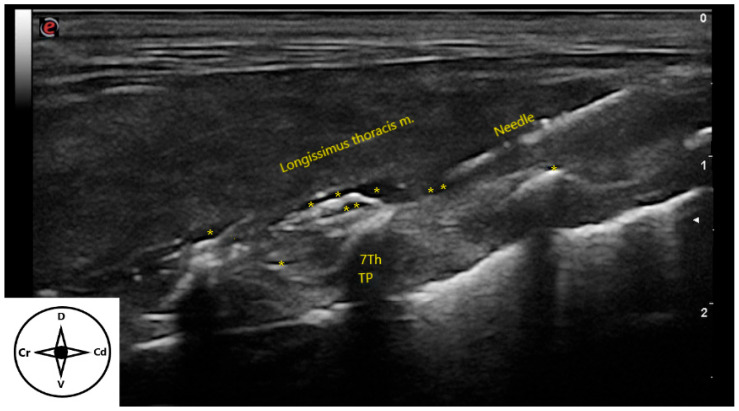
Ultrasonographic image of the distribution of injectate (*) after the longitudinal approach of the erector spinae plane block in a cat cadaver. The needle is inserted “in-plane” and advanced in a caudocranial direction until it contacts the transverse process of the seventh thoracic vertebra. TP: transverse process. Cr: cranial, Cd: caudal, D: dorsal, V: ventral.

**Figure 5 animals-15-03311-f005:**
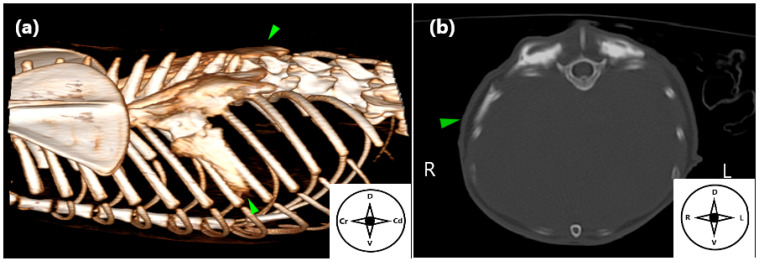
Computed tomography (CT) image illustrating the distribution of contrast medium following the longitudinal approach to the erector spinae plane block in a cat cadaver. (**a**) Three-dimensional CT reconstruction depicting contrast medium spread through the epaxial musculature and the seventh intercostal space (green arrowhead). (**b**) Transverse CT image confirming contrast distribution at the same level (green arrowhead). Cr: cranial, Cd: caudal, D: dorsal, V: ventral, R: right, L: left.

**Figure 6 animals-15-03311-f006:**
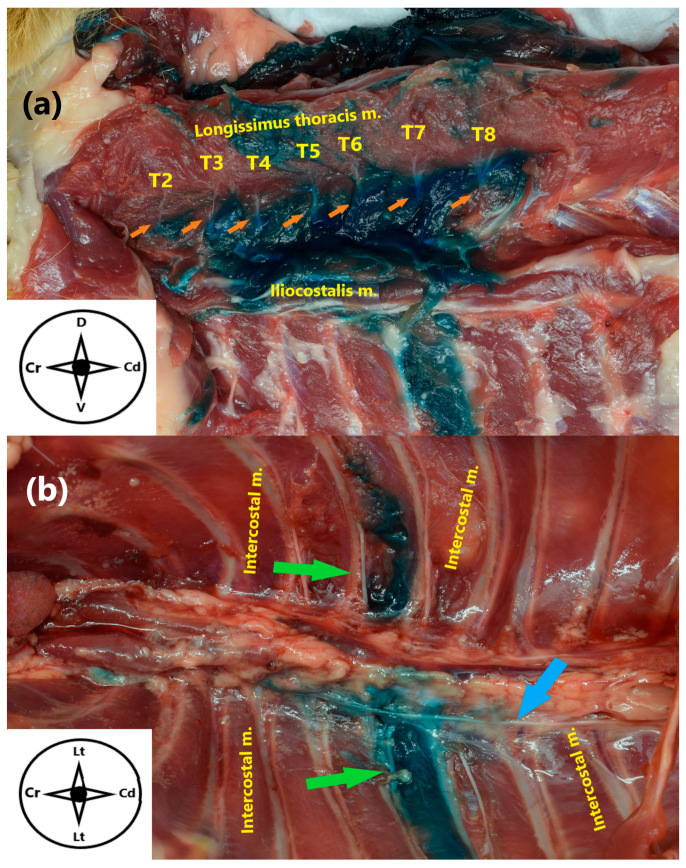
(**a**) Anatomical dissection of a cat cadaver in lateral recumbency, showing the intercostal and epaxial musculature of the spine following the longitudinal approach to the erector spinae plane block, showing stained dorsal branches from T2 to T8 (orange arrows). (**b**) Zenithal view of the inner dorsal thoracic wall after anatomical dissection, showing the stained ventral branch of T7 (green arrows) and the sympathetic trunk (blue arrow). Cr: cranial, Cd: caudal, D: dorsal, V: ventral, Lt: lateral.

**Figure 7 animals-15-03311-f007:**
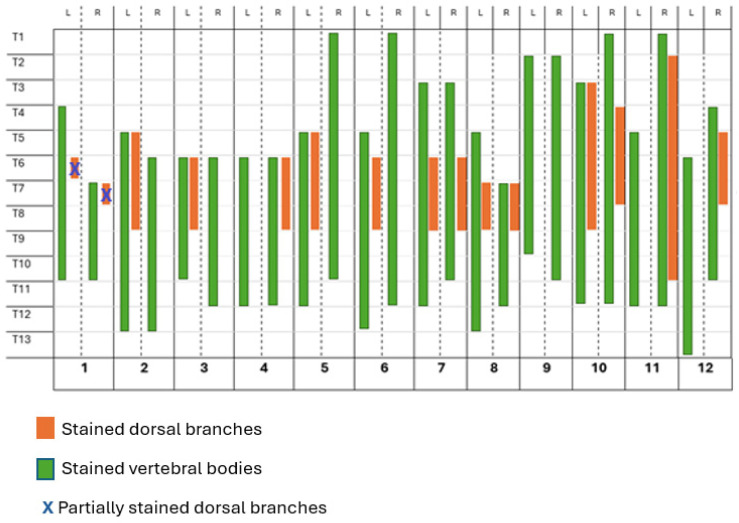
Number of vertebral bodies (CT distribution of contrast medium) and stained dorsal branches of the spinal nerves, following the longitudinal approach to the erector spinae plane block of a mixture of 0.4 mL kg^−1^ of methylene blue, lidocaine, and iopromide in 12 cat cadavers. L: left side; R: right side; T1–T13: spinal nerves and vertebral bodies; 1–12: cat cadavers.

## Data Availability

The original contributions presented in this study are included in the article. Further inquiries can be directed to the corresponding author.

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
