# Peer review of "Assessment of an Ultrasound-Guided Longitudinal Approach to the Thoracic Erector Spinae Plane Block in Cat Cadavers: Description of Dye and Contrast Medium Distribution"

_animals, 2025, doi:10.3390/ani15223311_

Round 1

Reviewer 1 Report

Comments and Suggestions for Authors

Dear authors,

Thank you for allowing me to review your manuscript "Assessment of an ultrasound-guided longitudinal approach to the thoracic erector spinae plane block in cat cadavers: description of dye and contrast medium distribution". It presents an original and well-structured study of erector spinae plane (ESP) block in cats, being the first study to describe the thoracic approach using ultrasound and confirm its distribution with computed tomography and anatomical dissection.

The experimental design is sound, the technical description is clear, and the results are consistent with previous findings in other species. The combination of imaging techniques and anatomical analysis provides considerable added value for the understanding of interfascial blocks in veterinary medicine.

Despite this, I would like to make some minor comments and suggestions:

  • Please clarify in the Methods section why the injections were performed at the level of T7. A brief justification—such as its anatomical accessibility, proximity to relevant dorsal branches, or alignment with previous studies in dogs (Portela et al., 2020)—would strengthen the methodological rationale.
  • Please, in Materials and Methods, make reference to the technique used by Portela et al (2020), detailing it and including that the ultrasound image of the armchair is sought as a realiable landmark, as later commented in the discussion.
  • It appears that the abbreviations “TVP” and “TP” have been used interchangeably; please use “TP (Transverse Process)” consistently throughout the text, replacing all instances of “TVP,” including in figure legends and the abbreviations section.
  • The terms “Craneal” appearing in several figure legends (lines 161, 170, 221, 236, 252, and Abbreviations) should be replaced with “Cranial”.
  • Additionally, replace “dying of the sympathetic trunk” with “staining of the sympathetic trunk” (line 54) to ensure accurate terminology.
  • Please, replace “carcass” with “cadaver”, since cadaver is the appropriate and respectful term for animal bodies used in scientific or anatomical research, while carcass generally refers to animal parts intended for consumption.
  • In line 330, please replace corpuses by cadavers.
  • Some references do not follow the required MDPI format; please revise them so that journal names appear in italics, volumes in bold, and page numbers in full (e.g., reference [5]:  Anaesth. Analg.46, 516–522 (2019)).

Reviewer 2 Report

Comments and Suggestions for Authors

This is a well written paper providing important information about the potential use of the thoracic ESP block in cats. Please see comments below:

Line 66: Instead of “Besides, regional anaesthesia has been associated with……..”, it would make more sense to say, “Regional anaesthesia has also been associated with…….”.

Line 97: Please replace “explaining” with “explain”.

Line 147 and 292: An image showing the “armchair shape” of the TVP of T7 maybe beneficial to the reader.

Line 177: Please replace “recumbence” with “recumbency”.

Line 224: Please describe the population of cats used as was done for the anatomical study.

Line 225: Please replace “de” with “the”.

Line 246, 255, 258, 259, 262: Please replace “x, y” with “x. y”.

Line 316: Please replace “The decision of….” with “The decision to….”.

Line 338: While the word “consonance” can mean agreement, it also means a form of rhyme involving the repetition of identical or similar consonants in neighbouring words. To avoid confusion for the reader it may be better to replace it with “agreement”.

Line 357: Please replace “corpuses” with “corpses”.

Line 378: Please say “only at T7” instead of “always in T7” as this better describes the results of the study.

Fig. 1, 2, 3, 4, 5: Please replace “craneal” with “cranial”.

Fig. 1: The directional compass is labelled “Lt” on both sides, is this correct? Should it dorsal (D) and ventral (V)? Also, if Lt is correct, it can’t be in two directions on the compass.

Fig. 3: Please remove “Lt: Lateral” from figure description.

Fig. 5(b): The directional compass is labelled “Lt” on both sides. Should this be right (R) and left (L) as seen in the actual CT image and in the figure description?

Fig. 6: Please add in description of Cr, Cd, D and V

Fig. 6(b): Do the authors mean to use “Lt” in the directional compass or should it be dorsal (D) and ventral (V). Also, if Lt is correct, it can’t be in two directions on the compass.

Fig. 7: What do the orange and green colours represent?

Reviewer 3 Report

Comments and Suggestions for Authors

Thank you for submitting the manuscript “Assessment of an ultrasound-guided longitudinal approach to the thoracic erector spinae plane block in cat cadavers: description of dye and contrast medium distribution.” It is a very interesting topic, and it is encouraging to start generating data and creating literature for cats as well.

In general, materials and methods as well as results are well presented and described. I would have probably added the sample numbers in materials and methods instead of the results. However, it can be argued either way so I would leave that to you to decide. The introduction could have some minor improvements to highlight the background of this block and the possible clinical relevance and importance. In case you can foresee some differences in cats and dogs that remain to be proven (or disproven) that could be added as well. The discussion needs a bit more work. It will be good to try following the order that you are actually presenting your results while you are discussing them. Also, avoid introducing or even reintroducing results and values in your discussion and focus on making hypotheses on your findings while comparing them with relevant literature. To some extent your discussion was like a literature review that is not the point of the discussion. You have also an extended part on the clinical applicability that we can argue that is not strictly relevant to your study at the moment. It can be definitely mentioned as further clinical studies are needed but I would avoid focusing on that.

Also, the figures would benefit from some extra explanation and labelling of the muscles.

Lastly, make sure you actually need all the references you are including. There is a recently published cadaveric study for lumbar space ESP in cats and it makes sense to have extended comparison of the two and they can be complementary anyway.

Overall, that’s a nice and very well needed study that could improve with a more focused discussion while highlighting the need for further clinical studies

Please find comments for each section:

Simple summary:

It is well written and descriptive enough section, I would only question whether it is a very ‘simple’ summary for lay audience. Consider to omit or simplify some anatomy details if you think you can omit some details.

Introduction: is well written with some important references. It would be nice if the clinical relevance of this block in cats is mentioned as well.

Line 66: “Additionally” instead of ‘Besides’

Line 69: “desensitisation” instead of “desensitising”

Line 70: “high” before safety

Lines 70-71: write “low complication risk” instead of “low risk of complications” instead of “low risk of associated complications”

Line 74: delete “Furthermore”

Line 79: please provide some further details on the location of the injection or the options that exist for the exact location.

Lien 80: please mention the structures that are blocked with this ESP technique.

Line 84: “surgery” instead of “surgical”

Line 84: please mention examples from procedures

Lines 87-90: these lines could be reworded because the meaning id quite confusing. Do you mean innervation instead of intervention?

Lines 93-94: considering there are not veterinary cadaveric countless studies, here they are quite few mentioned that the VB are dyed and it worth mentioning couple of studies that was not case.

Line 99: delete “another possibility is that”

Line 104: delete “first”

Line 106: delete “second”

Materials and Methods:

Anatomical study: I appreciate that this is an excellent and detailed work. It would have been extremely helpful for the readership if you could provide a diagram with the different muscle groups. I must admit I had to search some of the muscles in order to visualise them and make sure I follow your manuscript.

Also, could you clarify if you have used one animal for the dissections? I am aware that this is included in your results. However, it seems appropriate to include it in the M&Ms. Maybe at least consider it.

Ultrasound guided technique:

How many cadavers did you use initially?

Line 146: I am not sure you have explained before the TVP abbreviation

Line 148: would this be the needle size you recommend for cats?

Line 161: “cranial” instead “craneal”

Line 177: “recumbency” instead of “recumbence”. Also, the limbs were extended in which direction?

Lines 186-187: do you base this classification as stained or partially stained on previous studies or you can provide a reference ?

Figure 3: could you please include the names of the muscles that are included in this figure?

Line 225: do you here after the initial administration of the 0.1 ml or the 0.4 ml/kg?

Line 244: which were those 11 vertebral bodies?

Figure 5: would it be possible to have the contrast medium in different colour in figure 5a?

Figure 6: will also benefit from adding the names of the muscles

Discussion: It will be good your discussion to be organised following the way you presented your result. First the anatomical study, followed by the ultrasound guided technique. Also, I am sorry if I have missed but can you explain why you chose T7 as the injection point was only the fact that is roughly in the middle or there is another reason for this choice?

Lines 281-282: the last sentence should be re-written along the lines of: “The above indicates that the technique could be used …

Lines 285-287: Is not very clear why you are discussing the ultrasound probe here. It could be mentioned later but not necessarily in that detail unless is causing some kind of issue or previous studies in cats have used different probe

Line 300: add “comparative” before studies

Lines 306-308: it should be mentioned however that the fascia blocks are volume depended blocks and it is not unusual to use larger volume

Line 307: this reference is not relevant to this block but is a general pharmacology of local anaesthetics

Lines 319-322: the anatomical study, as discussed above, should have been discussed before the ultrasound study

Lines 337-338: have you performed a sample calculation?

Line 338: which results are you referring to?

Lines 339-341: please rewrite, is not very clear what you actually mean

Lines 346-347: avoid introducing results in the discussion

Line 354-356: please rewrite this part and introduce a hypothesis for the differences in the findings of your study compared with others.

Line 364: you could start a new paragraph before “ the mechanism”

Lines 364-366: since you have not performed a clinical study to evaluate pain, is not clear why you are discussing the analgesic effect of the block

Lines 387: sample size calculation?

Line 395: add “the” before the amount and the distance

Round 2

Reviewer 3 Report

Comments and Suggestions for Authors

I would like to thank the authors for submitting a revised and much improved version of their paper: “Assessment of an ultrasound-guided longitudinal approach to the thoracic erector spinae plane block in cat cadavers: description of dye and contrast medium distribution”.

The manuscript reads much better and thank you very much for your effort and corrections.

Please find some minor corrections and recommendations:

Line 96-97: it would be nice to name some of the areas that are innervated by the ventral branches of the spinal nerves.

Line 142: ‘consisted of’ instead of ‘consisted in’

Line 188: it is better to avoid saying “Few minutes” and you can start your sentence as “ A helical CT was performed 15-20 minutes following the injections…

Lines 200-202: could you please add a reference on when a nerve is considered ‘stained’

Figure 3 b: this needs to be better explained, since you are describing the epaxial muscles and you are labelling the intercostal muscles, and it can be quite confusing.

In your discussion, do you have any additional comments on the intramuscular distribution of the mixture?

Author Response

Review answers in Word.
